# Site-selective electrooxidation of methylarenes to aromatic acetals

Peng Xiong[1], Huai-Bo Zhao[1], Xue-Ting Fan[1], Liang-Hua Jie[1], Hao Long[1], Pin Xu[1], Zhan-Jiang Liu[1], Zheng-Jian Wu[1], Jun Cheng [1✉] & Hai-Chao Xu [1,2✉]

Aldehyde is one of most synthetically versatile functional groups and can participate in numerous chemical transformations. While a variety of simple aromatic aldehydes are commercially available, those with a more complex substitution pattern are often difficult to obtain. Benzylic oxygenation of methylarenes is a highly attractive method for aldehyde synthesis as the starting materials are easy to obtain and handle. However, regioselective oxidation of functionalized methylarenes, especially those that contain heterocyclic moieties, to aromatic aldehydes remains a significant challenge. Here we show an efficient electrochemical method that achieves site-selective electrooxidation of methyl benzoheterocycles to aromatic acetals without using chemical oxidants or transition-metal catalysts. The acetals can be converted to the corresponding aldehydes through hydrolysis in one-pot or in a separate step. The synthetic utility of our method is highlighted by its application to the efficient preparation of the antihypertensive drug telmisartan.

---

[1] State Key Laboratory of Physical Chemistry of Solid Surfaces, Innovative Collaboration Center of Chemistry for Energy Materials, and College of Chemistry and Chemical Engineering, Xiamen University, 361005 Xiamen, PR China. [2] Key Laboratory of Chemical Biology of Fujian Province, Xiamen University, 361005 Xiamen, PR China. ✉email: chengjun@xmu.edu.cn; haichao.xu@xmu.edu.cn

**B**enzylic oxygenation of alkylarenes provides crucial access to many industrial chemicals, such as terephthalic acid, phenol, and acetone, on a multimillion-ton scale[1]. Aldehyde is one of the most versatile synthetic handles and can be converted to numerous functionalities. As a result, aromatic aldehydes have been widely used in the manufacture of fine chemicals, nutraceuticals, performance materials, and pharmaceuticals. The oxygenation of methylarenes is a straightforward and attractive strategy for the preparation of aromatic aldehydes, especially considering that the starting materials are widely available and easy to handle. However, partial oxidation of methylarenes to aldehydes remains a largely unsolved challenge due to the strong propensity of product overoxidation under aerobic conditions (Fig. 1a)[2,3], and unsatisfactory chemo- and regioselectivity with substrates bearing multiple oxidizable C–H bonds and/or functionalities[4]. Despite these difficulties, oxygenation of simple methylarenes to aldehydes has been achieved using stoichiometric chemical oxidants such as *o*-iodoxybenzoic acid (IBX)[5], ceric ammonium nitrate (CAN)[6], pyridinium chlorochromate[7] or polyoxometalate $H_5PV_2Mo_{10}O_{40}$[8]. Transition metal-catalyzed aerobic oxidation using hexafluoro-2-propanol as solvent[9] or by adding polymethylhydrosiloxane as reagents to avoid overoxidation[10] have also been reported (Fig. 1a). As an alternative to chemical oxidation, electrooxidation eliminates the use of stoichiometric chemical oxidants and is attracting increasing interests[11–26]. Notably, electrooxidation of electron-rich toluene derivatives to substituted benzaldehydes has been applied in the industrial production of *p*-anisaldehyde and 3,4,5-trimethoxybenzaldehyde[27–29]. Despite these accomplishments, the conversion of structurally complex methylarenes, including medicinally relevant benzoheterocycles in particular, has remained synthetically elusive because of selectivity issues and catalyst inhibition by the coordinating heteroatoms[30].

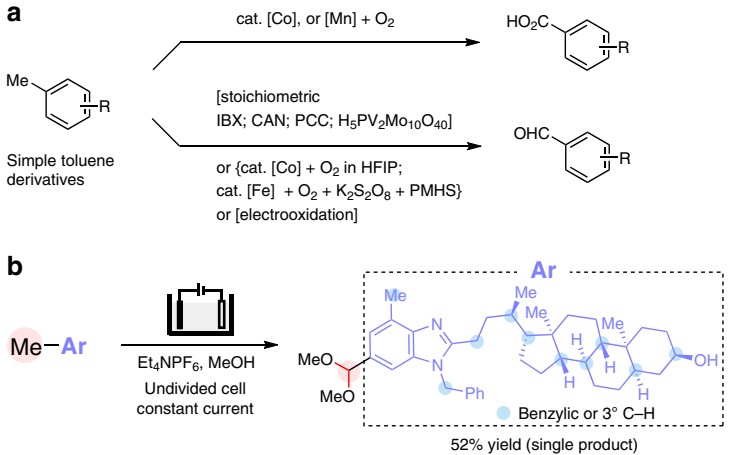

**Fig. 1 Methylarene oxidation. a** Examples of reported oxidation of relatively simple toluene derivatives to benzoic acids or benzaldehydes. **b** The present study focuses on site-selective electrooxidation of methyl benzoheterocycles to aromatic aldehydes.

**Table 1 Optimization of reaction conditions[a].**

| Entry | Deviation from standard conditions | Yield (%)[b] |
|---|---|---|
| 1 | None | 72[c] |
| 2 | Reaction at RT | 25 |
| 3 | Reaction at 8 mA | 61 |
| 4 | Reaction at 12 mA | 66 |
| 5 | Reaction under air | 50 |
| 6 | $Et_4NPF_6$ (0.2 equiv) | 63 |
| 7 | $Et_4NBF_4$ instead of $Et_4NPF_6$ | 72 |
| 8 | *n*-$Bu_4NPF_6$ instead of $Et_4NPF_6$ | 64 |
| 9 | $Et_4NOTs$ instead of $Et_4NPF_6$ | 66 |
| 10 | Pt plate (1 cm × 1 cm) as anode | 20 (69)[d] |
| 11 | Graphite plate (1 cm × 1 cm) as anode | 56 |

*Bn* benzyl
[a]Reaction conditions: RVC anode, Pt plate cathode, **1** (0.2 mmol), MeOH (9 mL), $Et_4NPF_6$ (0.1 mmol), 10 mA, 2.3 h (4.3 F mol$^{-1}$).
[b]Determined by $^1$H NMR analysis using 1,3,5-trimethoxybenzene as the internal standard.
[c]Isolated yield.
[d]Unreacted **1** in bracket.

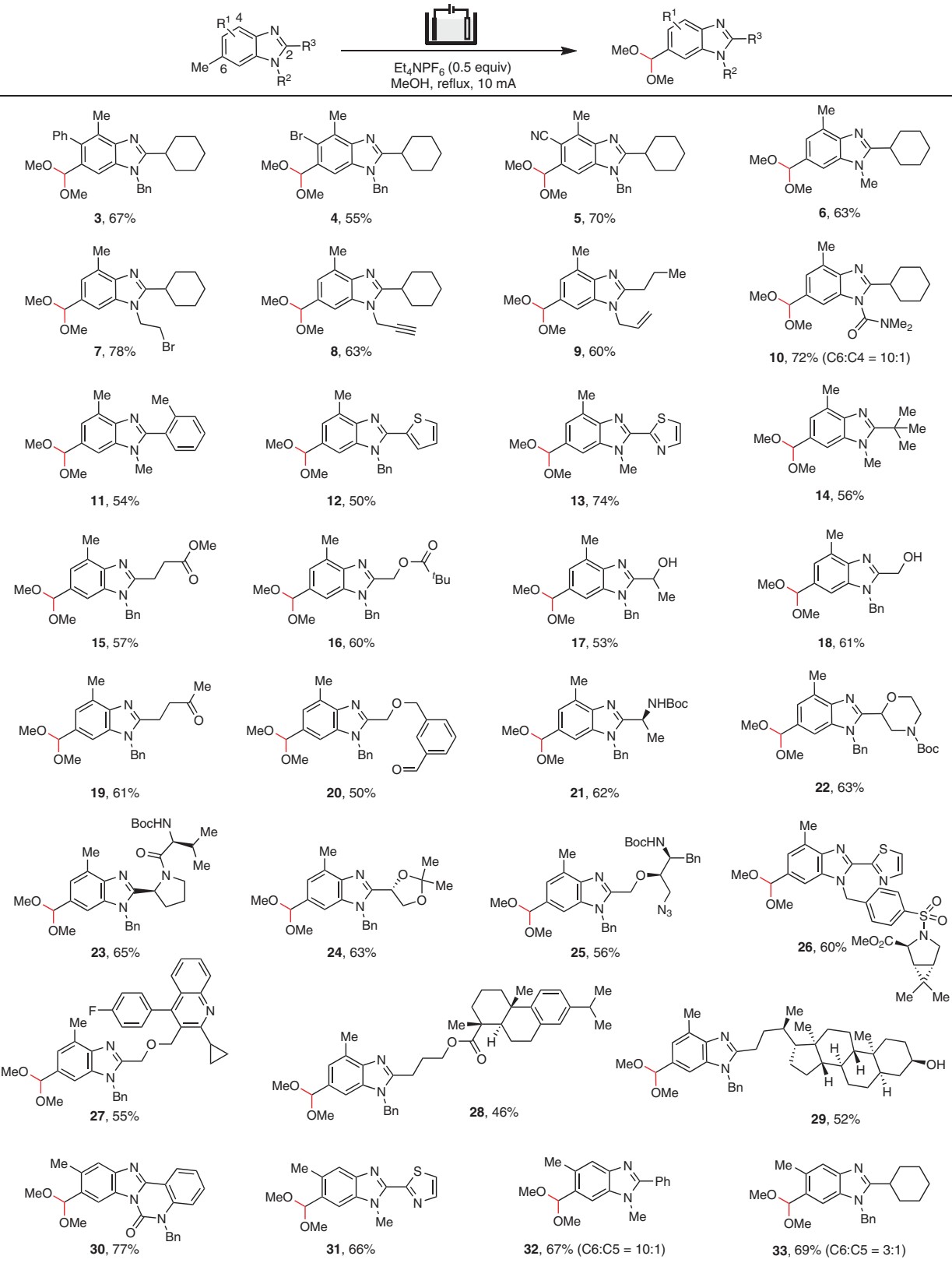

**Fig. 2 Scope of site-selective electrooxidation of benzimidazoles.** Reaction conditions: heterocycle (0.2 mmol), MeOH (0.022 M), reflux, 2.2–4.5 h. All yields are isolated yields. Regioisomers were not observed unless otherwise mentioned.

We have been interested in electrochemical synthesis of heterocycles[23,31,32] and recently reported intramolecular dehydrogenative cyclization reactions for the preparation of several types of benzoheterocycles[33–37]. Alternatively, we envision the synthesis of functionalized benzoheterocycles by modification of alkyl side chains of existing benzoheterocyclic scaffolds. Herein we report a generally applicable electrochemical strategy capable of oxidizing various methyl benzoheterocycles to aromatic acetals in a site-selective manner (Fig. 1b). These side chain oxidation reactions allow access to various functionalized benzoheterocycles difficult to obtain directly through cyclization processes.

## Results

**Reaction optimization.** We began our study by first optimizing the electrooxidation of benzimidazole **1** bearing Me groups at positions 4 and 6 (Table 1). The best results were achieved in an undivided cell with refluxing methanol as solvent, Et$_4$NPF$_6$ (0.5 equiv) as electrolyte, a Pt plate cathode, and a reticulated vitreous carbon anode. Under these conditions, compound **1** reacted site-selectively at C6-Me group to give dimethyl acetal

**2** in 72% yield (entry 1) without overoxidation to orthoester or unwanted oxidation of other potentially labile substituents such as C4-Me, N–Bn, or 3° C–H on the cyclohexyl moiety. Lowering the reaction temperature to RT dramatically decreased the yield of **2** to 25% (entry 2). Furthermore, moderate reduction in reaction efficiency was observed when the electrolysis of **1** was performed at a different current (entries 3 and 4), under air (entry 5), with a decreased amount of Et$_4$NPF$_6$ (entry 6), or with another electrolyte such as $n$-Bu$_4$NPF$_6$ (entry 8) or Et$_4$NOTs (entry 9). Et$_4$NBF$_4$ (entry 7) was, however, equally effective as a supporting electrolyte. Product formation was also diminished with the use of a Pt (entry 10) or graphite anode (entry 11).

**Evaluation of substrate scope.** With the optimized reaction conditions defined, we set out to explore the scope of the electrooxidation of methylarenes. Notably, the site-selectivity was not significantly affected by introducing a phenyl (**3**), bromo (**4**), or cyano group (**5**) at C5, or by varying the substituent on N1 (**6–9**) or C2 (**11–29**) of the C4,C6-dimethylated benzimidazole substrate (Fig. 2). However, the installation of a carbamoyl group on N1

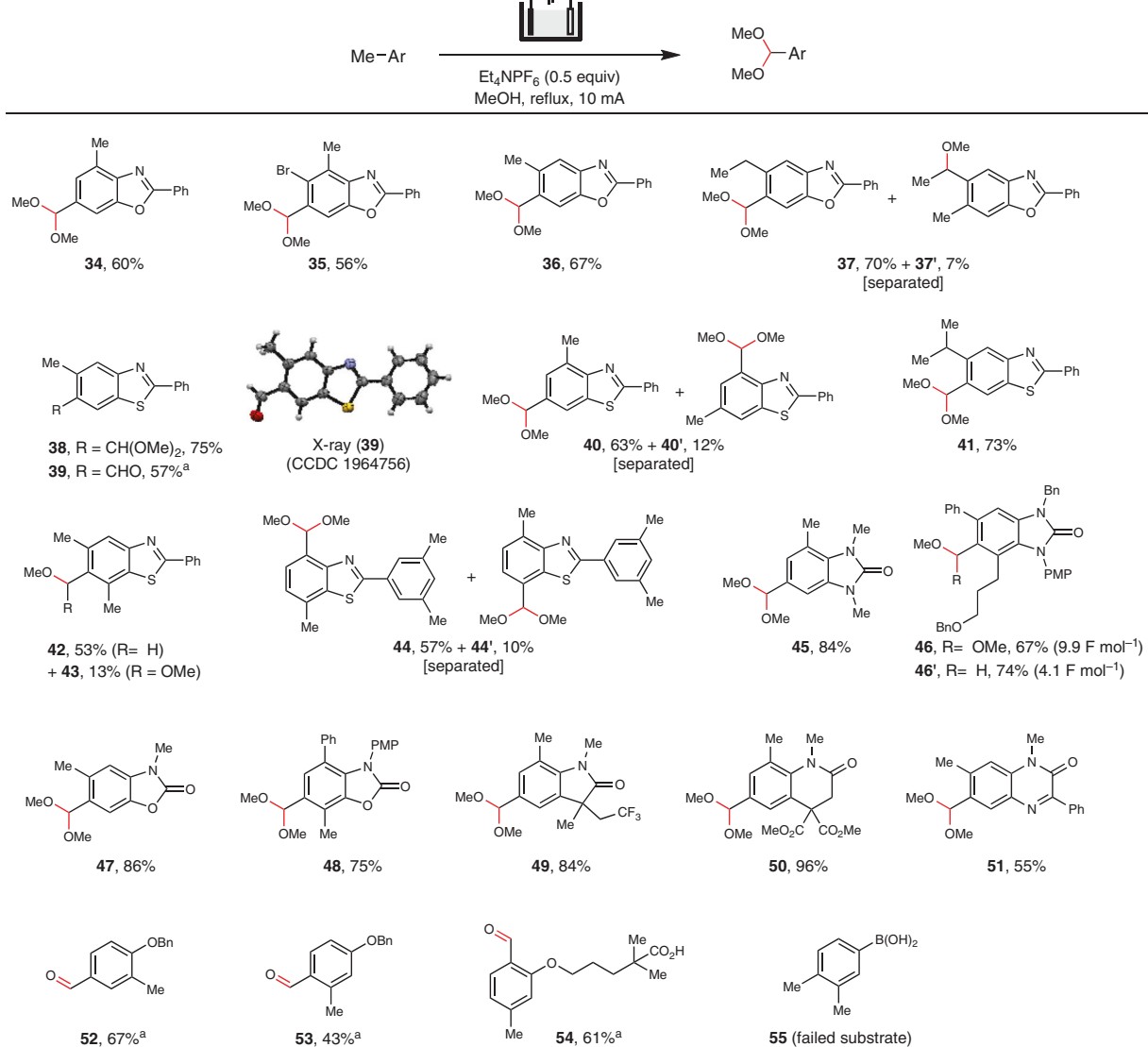

**Fig. 3 Electrooxidation of various methyl benzoheterocycles.** Reaction conditions: methylarene (0.2 mmol), MeOH (0.022 M), reflux, 2.2–5.3 h. All yields are isolated yields. Regioisomers were not observed unless otherwise mentioned. [a]Electrolysis was followed by hydrolysis with aqueous HCl (2 N).

resulted in a slight decrease of regioselectivity (**10**). The method showed broad compatibility with common functional groups or moieties such as alkyl bromide (**7**), alkyne (**8**), alkene (**9**), ester (**15**, **16**), alcohol (**17**, **18**), ketone (**19**), aldehyde (**20**), Boc-protected amine (**21–23**), ketal (**24**), azido (**25**), and aromatic heterocycles (**12**, **13**, **26**, and **27**). Molecular fragments derived from natural products dehydroabietic acid (**28**) and lithocholic acid (**29**) were equally well tolerated. On the other hand, site-selective oxidation of the C6-Me group in C5,C6-dimethylated benzimidazoles bearing an aryl substituent at C2 (**30–32**) could also be achieved. The replacement of the aryl group with a cyclohexyl, however, resulted in a moderate site-selectivity (**33**). This reduction in site-selectivity for the 2-cyclohexyl substrate was probably caused by the increased reactivity of the corresponding radical cation compared with the 2-aryl counterparts.

Benzoxazoles (**34–37**) and benzothiazoles (**38**, **40–42**) with multiple open benzylic positions were all found to undergo site-selective oxidation at the C6-Me group (Fig. 3). The site-selectivity was maintained even for substrates bearing an ethyl (**37**) or isopropyl group (**41**) at C5 that contained secondary or tertiary benzylic C–H bonds. Notably, the oxidation of 5,6,7-trimethyl benzothiazole proceeded site-selectively as intended despite the high steric hindrance of its C6-Me substituent. However, the resultant product mixture comprised mono-methoxylated **42** as the main product with a minor amount of acetal **43**, because the steric environment was detrimental to the second C–H cleavage[38]. Meanwhile, oxidation of a C4,C7-dimethylated benzothiazole with a methylated phenyl group on C2 occurred preferentially on the C4-Me (**44**). The electrooxidation method was successfully extended to many other benzoheterocycles, including 2-benzimidazolidinone (**45**, **46**), 2-benzoxazolone (**47**, **48**), 2-oxindole (**49**), 3,4-dihydro-1H-quinolin-2-one (**50**), and quinoxalinone (**51**). Once again, probably due to the steric hindrance, monomethoxylated product **46′** could be obtained selectively with good yield when the electrolysis was stopped at 4.1 F mol$^{-1}$. The electrochemical method was not limited to benzoheterocycles as demonstrated by the site-selective oxidation of methylated alkoxybenzenes (**52–54**). The relatively electron-deficient 3,4-dimethylphenylboronic acid (**55**), however, decomposed into intractable material and did not afforded any aldehyde product. The above results clearly suggested that the site-selectivity for the electrochemical benzylic oxidation reaction are not controlled by steric effects or bond dissociate energies (BDEs) of the C–H bonds.

The Me oxidation reaction could be coupled with amidine cyclization that we previously described to construct functionalized benzimidazoles (**56–65**) and imidazopyridines (**66–70**) (Fig. 4)[33]. The reaction of an amidine containing a 3,4-disubstituted phenyl ring afforded two products **62** and **62′** because of unselective cyclization. The benzylic oxidation was, however, selective for both regioisomers. Compound **71** did not undergo further Me oxidation because its oxidation potential exceeded the decomposition potential of MeOH solvent. The tandem cyclization/Me oxidation process provided access to benzimidazoles with substitution patterns difficult for the existing methods[33,39].

**Synthesis of telmisartan.** The synthetic utility of our electrooxidation reaction was demonstrated through the construction of

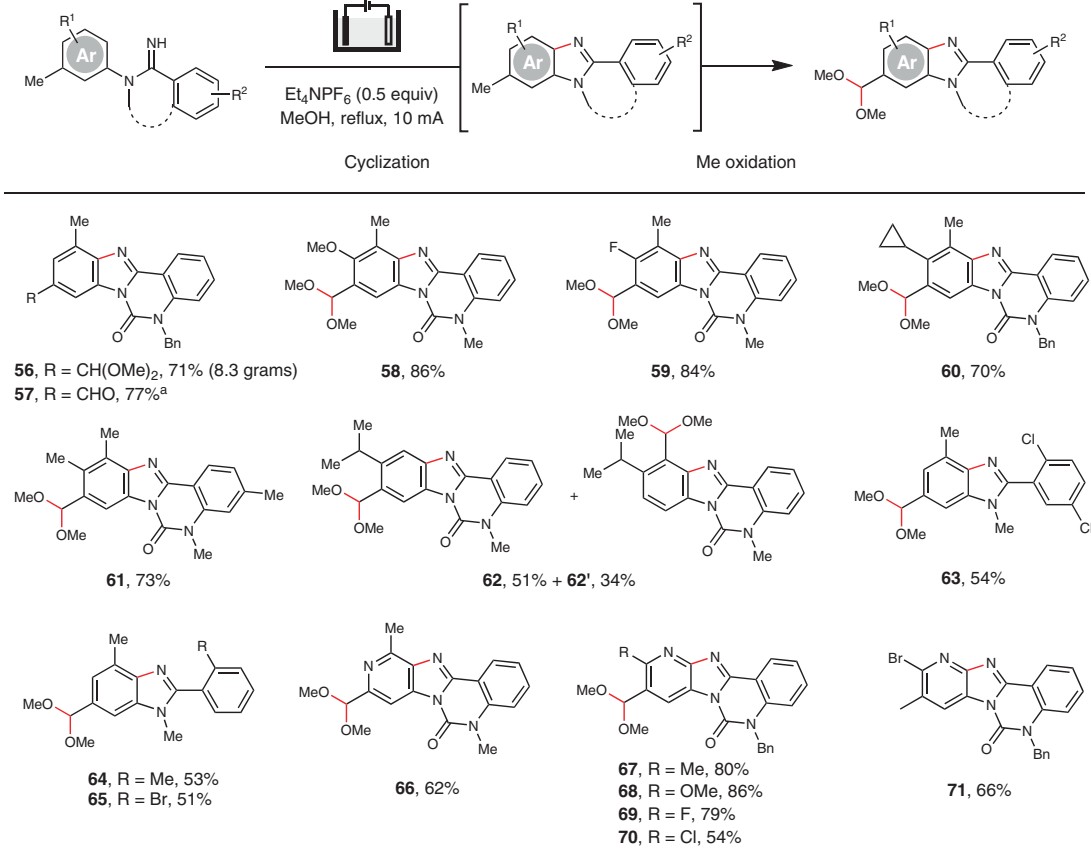

**Fig. 4 Electrochemical cyclization/benzylic oxidation of amidines.** General reaction conditions: amidine (0.2 mmol), MeOH (0.022 M), reflux, 3.5–12.5 h. All yields are isolated yields. Regioisomers were not observed unless otherwise mentioned. [a]Electrolysis was followed by one-pot hydrolysis with HCl (2 N).

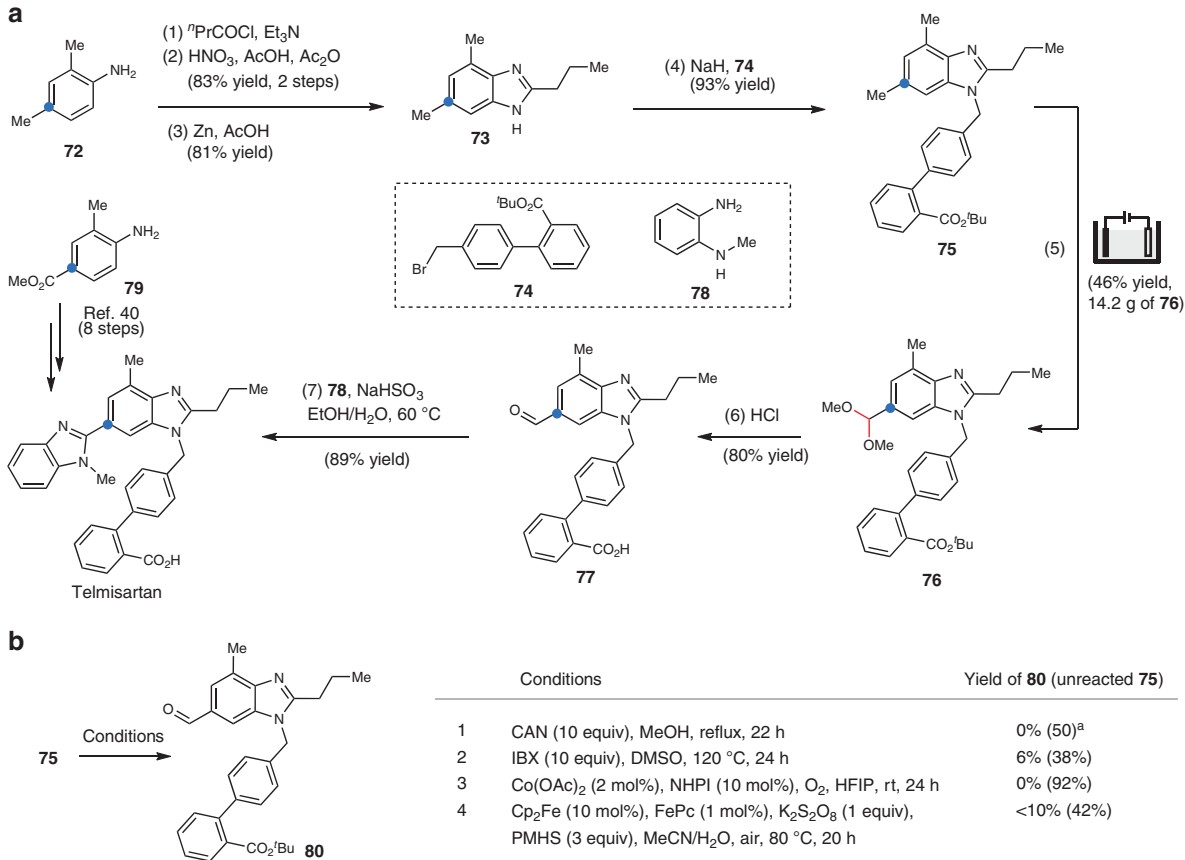

**Fig. 5 Synthesis of telmisartan. a** Our synthetic route employing site-selective benzylic electrooxidation as a key step. **b** Oxidation of **75** under reported conditions for benzylic oxidation. [a]Me ester was recovered instead of the original [t]Bu ester **75** because of transesterification. NHPI N-Hydroxyphthalimide, Fe(II)Pc iron(II) phthalocyanine.

the antihypertensive drug telmisartan (Fig. 5a). We first prepared benzimidazole **75** from a commercially available aniline **72** in four steps. Subsequently, site-selective electrooxidation of **75** afforded dimethyl acetal **76** in 46% yield on a decagram scale. In contrast, the oxidation of **75** by a stoichiometric amount of chemical oxidant such as CAN[6] or IBX[5], in the presence of a Co catalyst under aerobic conditions[9], or iron catalyzed oxidation[10] with $K_2S_2O_8$ afforded **80** in <10% yield despite the success of these methods with toluene derivatives (Fig. 5b). Compound **76** was then converted to telmisartan by treating with aqueous HCl to hydrolyze its acetal group to aldehyde and remove its [t]Bu, followed by condensation with o-phenylenediamine **78**. Notably, the starting material **72** employed in this synthetic route are much less expensive than ester **79** used in a previously published 8-step method[40].

**Mechanistic investigation**. The reaction regioselectivity that we observed in this study suggested that the mechanism likely involved single electron transfer oxidation of the benzene nucleus to a radical cation, followed by benzylic C–H cleavage[28]. This hypothesis is further supported by the finding that bromination of benzoxazole **81** with NBS, known to proceed through hydrogen atom transfer, afforded a regioisomeric mixture of **82** (50%) and **83** (17%) along with dibrominated **84** (10%) (Fig. 6a). Density functional theory calculations were also performed to provide a plausible rationale for the origin of the observed site-selectivity (Fig. 6b). We first analyzed the distributions of the lowest unoccupied molecular orbitals (LUMO) of the radical

cations derived from benzimidazoles (**I–III**), benzoxazoles (**IV**, **V**), benzothiazoles (**VI**, **VII**), and 2-benzoxazolone (**VIII**) that bear multiple Me groups. As shown in Fig. 6b, the LUMOs are delocalized throughout the carbon skeletons of the benzoheterocycles with the distributions on C6 atoms being higher than other carbon atoms attached with a Me group. Furthermore, the natural population analysis shows that the charges of C6 are also more positive than those of other atoms bearing a Me group, indicating deprotonation of the C6-Me groups is preferred[41].

In summary, we have shown that electrooxidation of methyl benzoheterocycles occurs in a site-selective manner to afford a wide range of structurally diverse aromatic acetals. The site-selectivity is governed by the innate electronic properties of the benzo ring instead of BDEs of the C(sp³)–H bonds. The benzylic oxidation takes place efficiently in a simple undivided cell and employs traceless electric current as the reagents without need for stoichiometric chemical oxidants. These features render the reactions scalable and attractive for industrial scale applications.

## Methods
**Representative procedure for the electrooxidation of methylarenes**. A 10 mL three-necked round-bottomed flask was charged with **1** (0.20 mmol, 1.0 equiv) and $Et_4NPF_6$ (0.10 mmol, 0.5 equiv). The flask was then equipped with a condenser, a reticulated vitreous carbon (100 PPI, 1.2 cm × 1.0 cm × 0.8 cm) anode and a platinum plate (1.0 cm × 1.0 cm) cathode, and flushed with argon. MeOH (9.0 mL) was then added. The electrolysis was carried out at 80 °C (oil bath temperature) using a constant current of 10 mA until complete consumption of the substrate (2.3 h, 4.3 F mol⁻¹). The reaction mixture was cooled to RT and concentrated under reduced pressure. The residue was chromatographed through silica gel eluting with ethyl acetate/hexanes containing 1% triethylamine to give the desired product **2** in

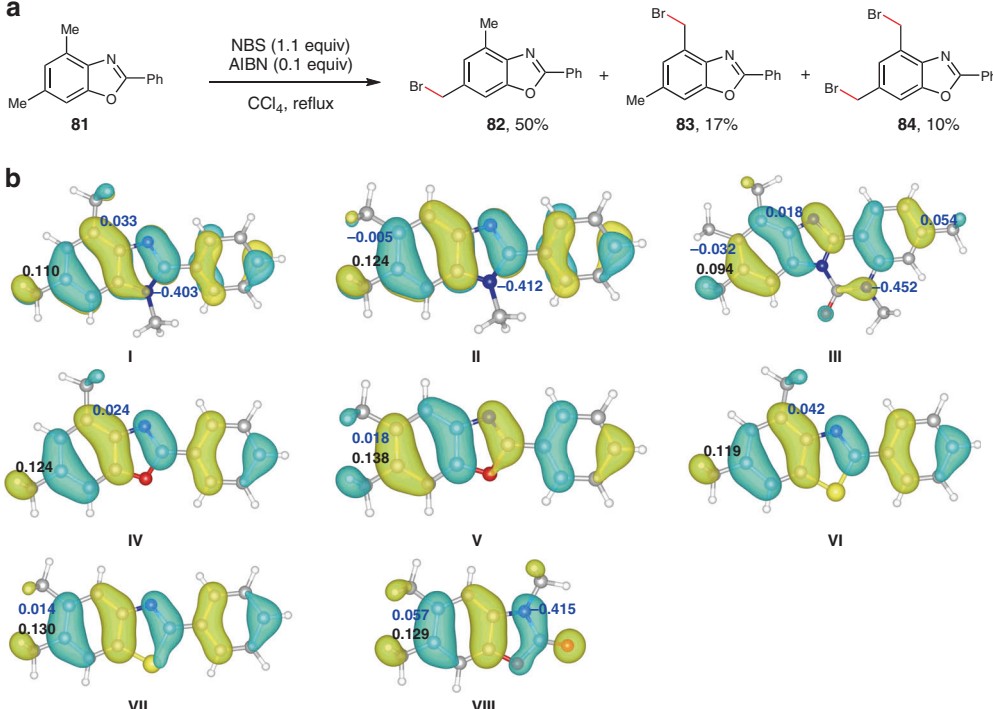

**Fig. 6 Investigation on the site-selectivity of the electrooxidation of methylarenes. a** Radical bromination of **81** with NBS. **b** Computed LUMOs and NPA charges of radical cations derived from various methylated benzoheterocycles. N, C, O, S, and H atoms are colored in blue, gray, red, yellow and white, respectively. The LUMOs are visualized by light blue and yellow isosurfaces. The NPA charges of the atoms bearing a Me group are indicated by black (C6) and blue numbers. NBS N-bromosuccinimide, AIBN azobisisobutyronitrile.

72% yield as a white solid. All new compounds were fully characterized (See the Supplementary methods).

## Data availability

The X-ray crystallographic coordinates for structures reported in this study have been deposited at the Cambridge Crystallographic Data Centre (CCDC), under deposition number 1964756. These data can be obtained free of charge from The Cambridge Crystallographic Data Centre via www.ccdc.cam.ac.uk/data_request/cif. The data supporting the findings of this study are available within the article and its Supplementary Information files. Any further relevant data are available from the authors on request.

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

## Acknowledgements
The authors acknowledge the financial support of this research from MOST (2016YFA0204100), NSFC (No. 21672178, 21971213), and Fundamental Research Funds for the Central Universities.

## Author contributions
P.Xiong and H.B.Z. contributed equally to this work. P.Xiong, H.B.Z., L.H.J., H.L., P.Xu, Z.J.L., and Z.J.W. performed the experiments and analyzed the data, X.T.F. and J.C. conducted the theoretical studies. H.C.X. designed and directed the project and wrote the manuscript.

## Competing interests
The authors declare no competing interests.
