## [Peer Review File · Nature Communications]

Reviewers' comments:

Reviewer #1 (Remarks to the Author):

The authors describe a new electrochemical strategy for the oxidation of methyl benzoheterocycles to aromatic aldehydes in a site-selective manner. The electrochemical tandem reactions of amidines cyclization and benzylic C-H oxidation were also applicable to obtain medicinally relevant benzoheterocycles. The synthetic utility of this electrochemical protocol was further demonstrated by the total synthesis of antihypertensive drug telmisartan. The present electrochemical strategy led to new solutions to challenging site-selective benzylic C-H oxidations that are not readily addressed using existing tools. Considering the novelty of this work and synthetic utility of this method to obtain medicinally relevant benzoheterocycles, I suggest its publication in Nat. Commun.. To improve the quality of the manuscript, the following minor issues should be addressed.

1) Fig. 2, the replacement of C2-Ph group with a cyclohexyl led to the sharp decrease of the site-selectivity. The reason is not completely clear to me. An explanation for the poor site-selectivity in product 33 is suggested to be added in the revised manuscript.

2) The mechanism was proposed involving SET oxidation of the benzene nucleus followed by benzylic C-H cleavage. Base additives may be beneficial for this process. Besides, the successive electrochemical oxidation steps should be given in the Supporting Information for readers to better understand the mechanism.

Reviewer #2 (Remarks to the Author):

In this manuscript, Xu and coworkers described an efficient electrochemical oxidation of methylarenes to aromatic aldehydes. This method essentially enables chemists to use methyl group — ubiquitous and hardly reactive — as a protected form of aldehyde. Accordingly, strategic use of this reaction will allow chemists to access important intermediates much more concisely from considerably cheaper starting material, as demonstrated in the synthesis of Telmisartan in figure 5. However, critical drawbacks of this reaction is its limitation of substrates to benzoheterocycles and substrate-controlled selectivity (to be fair, this is advantageous for target-oriented synthesis, but disadvantageous for diversity-oriented synthesis), which was briefly addressed in the abstract of this manuscript. I would also like to comment on novelty aspect of this work. As already recognised by the authors, electrochemical oxidation of methylarenes to benzaldehyde derivatives is a well-established reaction and has been practiced in industrial scale (ref 27-29 in the manuscript). The conditions reported here are essentially identical to those already known. Therefore, I see the value of this work is not the novelty of reaction, but rather synthetic utility of this transformation in the context of more complex molecule synthesis. Then, my concern would be that the readers are intrinsically limited to synthesis-oriented chemists. Based on this lack of novelty and broader interest, I think this work is more suited to be published in specialised journals such as Organic Letters or Journal of Organic Chemistry.

Specific comments and questions are as follows:

1) Product 10 shows slight decrease in regioselectivity. How would regioselectivity be changed if stronger electron-withdrawing group such as tosyl or nosyl is attached? If the regioselectivity further decreases, it strengthens the authors' arguments that regioselectivity of the oxidation is controlled by electronic environment of a substrate.

2) Yields are usually moderate (50-70%) .What are the rest of the material?

3) I don't fully understand the purpose of NBS bromination (figure 6a). Do the authors want to claim that mechanism is likely different based on the observation of regioisomers in the case of bromination?

4) Analysis of LUMO of radical cation to account for the regioselectivity makes sense, but I found it

is not very intuitive. Do you think we can draw the same conclusion by using the stability of benzylic radicals, instead of LUMO of radical cation?

Reviewer #3 (Remarks to the Author):

In this communication, Cheng, Xu and co-workers reported a method for site-selective electrochemical oxidation. The benzylic methyl C-H bond is cleaved via anodically oxidation to afford the coupling product. The structure of the synthetic products were confirmed through X-ray single crystal analysis. The site-selective product may be subjected to hydrolysis, affording the corresponding aldehydes, which could be converted into the antihypertensive drug telmisartan. Additional DFT calculations have been carried out to explore the detailed mechanism of this transformation. The reaction has been demonstrated on various heterocycles with excellent yields and excellent site-selectivity. However, the manuscript and the supplementary information lack any detailed disclosure on the proposal mechanistic cycle. This should be remedied if possible. Overall, this transformation represents a useful method to access aromatic aldehydes from methylarenes. The substrates scope was broad with great regiocontrol. The broad selection of substrates may elicit general interest in the community. Taken together, this manuscript does possess the broad impact expected of a Nat. Comm. Publication, but some revisions are necessary prior to publication (vide supra):

1. The scope of the reaction has been well demonstrated by the authors. Under optimized conditions, substrates reacted site-selectively at C6-Me group to give dimethyl acetal. However, in some case, site-selective oxidation of C4-Me group could also be achieved (40'). How about the other kinds of the substrate only with C4-Me group? This is a minor point, but if the authors have already attempted other substrates, it's better to include them, even if the results are not good.
2. Aside from the foregoing, the reviewer has some concerns over the title of the manuscript. A wide range of structurally diverse aromatic dimethyl acetal derivatives has been disclosed via the electrooxidation of methyl heterocycles in a site-selective manner. However, the dimethyl acetals were isolated as the products. There is inaccurate in highlighting aldehyde in the title.

Reviewer #4 (Remarks to the Author):

The paper entitled "Site-selective electrooxidation of methylarenes to aromatic aldehydes" focuses on the synthesis of acetals starting from methylarenes using electrochemical tools to be further used for aldehyde synthesis approaches. This work is an ongoing research in which the authors previously reported an electrochemical synthesis of functionalized benzimidazoles and imidazopyridines via amidine cyclization. For this new manuscript the authors obtained a very completed and exhaustive library which is quite well characterized with the suitable techniques for the purpose. The compounds were produced in an eco-friendly mode coupled with good side-selectivity and yields which is crucial for future scale-up prospective. Therefore, the following minor remarks must be addressed by the authors before this paper could be published in "Nature Communications" journal.

1. In fact, the authors synthesize acetals employing electrochemical tools with methylarenes as starting materials. However, the aromatic aldehydes were obtained under further hydrolysis of these acetals. Therefore, this article title is misleading, while someone is trying to search for a paper, the first impression this title causes it that the authors obtained aromatic aldehydes from electrooxidation techniques, which is not completely true (just the previous step, or two steps before the hydrolysis that give rise to the final aldehydes). In contrast, other papers such as J. Electroanal. Chem. 751 (2015) 105–110, which is entitled "A promising electro-oxidation of

methyl-substituted aromatic compounds to aldehydes in aqueous imidazole ionic liquid solutions” is an example where the authors did electrochemical synthesis from methyl-substituted compounds to obtain the aldehydes as final products. Also, sentences such as referred in page 2: “Herein we report a generally applicable electrochemical strategy capable of oxidizing various methyl benzoheterocycles to aromatic aldehydes in a site-selective manner” shall be modified accordingly along the manuscript.

2. At the end of introduction the authors shall insert a paragraph telling the methylarenes choice (you can also highlight that this is an ongoing research by the group in which previous steps like the benzimidazoles electrosynthesis via amidine cyclization were performed with success) and also that main focus is targeting compounds like the antihypertensive drug telmisartan in an eco-friendly path. So that, it is easier for the reader to follow the main aim of this research.

3. After table 1, the authors are not talking about the reaction conditions anymore and thereafter a simple sentence must be inserted here to separate the reaction conditions optimization “section” from the use of compounds with different substituents, so as to the readers can follow better the logical reasoning applied.

4. No efficiency reduction was observed from entry 1 to entry 7 (Table 1), as it is referred in page 3, line 3, and therefore must be excluded from that sentence.

5. Page 5, I agree with last sentence where the authors say that site-selectivity is not controlled by steric effects. However, the authors refer some lines above that “...resultant product mixture comprised mono-methoxylated 42 as the main product with a minor amount of acetal 43, because the steric environment was detrimental to the second C–H cleavage”, if so, how do you explain the contrast obtained for compound 46? The text must be modified accordingly.

6. Conclusions shall be improved, this article has impact and will get attention from several readers even more in a journal as “Nature Communications”. The manuscript contains important matter for a scale-up prospective and I suggest the authors to refer something related with an industrial future point of view in order to avoid at a large scale the use of chemical oxidants and so on, as referred in the introduction.

Point-by-point response to the referees' comments is as follows:

Reviewer #1 (Remarks to the Author):

The authors describe a new electrochemical strategy for the oxidization of methyl benzoheterocycles to aromatic aldehydes in a site-selective manner. The electrochemical tandem reactions of amidines cyclization and benzylic C-H oxidation were also applicable to obtain medicinally relevant benzoheterocycles. The synthetic utility of this electrochemical protocol was further demonstrated by the total synthesis of antihypertensive drug telmisartan. The present electrochemical strategy led to new solutions to challenging site-selective benzylic C-H oxidations that are not readily addressed using existing tools. Considering the novelty of this work and synthetic utility of this method to obtain medicinally relevant benzoheterocycles, I suggest its publication in Nat. Commun.. To improve the quality of the manuscript, the following minor issues should be addressed.

Response: We thank the reviewer for the favorable recommendation.

1) Fig. 2, the replacement of C2-Ph group with a cyclohexyl led to the sharp decrease of the site-selectivity. The reason is not completely clear to me. An explanation for the poor site-selectivity in product 33 is suggested to be added in the revised manuscript.

Response: We thank the reviewer for this suggestion. We have added the following discussion to the manuscript: This reduction in site-selectivity for the 2-cyclohexyl substrate was probably caused by the increased reactivity of the corresponding radical cation compared with the 2-aryl counterparts. The selectivity determining step is the deprotonation of the arene radical cation. Since the methyl groups at C5 and C6 are of similar steric properties, electronic effects are most likely responsible for the selectivity. As we can see from Fig 6, LUMO and NPA charges are partially distributed to the 2-aryl group, suggesting that the 2-aryl group can help stabilizing the radical cation. Hence radical cation derived from 2-aryl substrate is expected to be less reactive than that from 2-alkyl substrate. The more reactive 2-alkyl substrate derived radical cation is thus less selective.

2) The mechanism was proposed involving SET oxidation of the benzene nucleus followed by benzylic C-H cleavage. Base additives may be beneficial for this process. Besides, the successive electrochemical oxidation steps should be given in the Supporting Information for readers to better understand the mechanism.

Response: We thank the reviewer for this suggestion. We have added the mechanism to the Supporting Information. Base additives are not beneficial for the reaction. For

example, the electrolysis of compound 66 in the presence of 2 equiv of pyridine led to a reduction in yield to 23% from 46% without a base. The radical cations are highly acidic (pKa of toluene radical cation: -13 in MeCN) and their deprotonation are not difficult. In addition, the use of bases may lead to reduced selectivity and facilitate oxidative decomposition of MeOH solvent.

Reviewer #2 (Remarks to the Author):

In this manuscript, Xu and coworkers described an efficient electrochemical oxidation of methylarenes to aromatic aldehydes. This method essentially enables chemists to use methyl group — ubiquitous and hardly reactive — as a protected form of aldehyde. Accordingly, strategic use of this reaction will allow chemists to access important intermediates much more concisely from considerably cheaper starting material, as demonstrated in the synthesis of Telmisartan in figure 5. However, critical drawbacks of this reaction is its limitation of substrates to benzoheterocycles and substrate-controlled selectivity (to be fair, this is advantageous for target-oriented synthesis, but disadvantageous for diversity-oriented synthesis), which was briefly addressed in the abstract of this manuscript. I would also like to comment on novelty aspect of this work. As already recognised by the authors, electrochemical oxidation of methylarenes to benzaldehyde derivatives is a well-established reaction and has been practiced in industrial scale (ref 27-29 in the manuscript). The conditions reported here are essentially identical to those already known. Therefore, I see the value of this work is not the novelty of reaction, but rather synthetic utility of this transformation in the context of more complex molecule synthesis. Then, my concern would be that the readers are intrinsically limited to synthesis-oriented chemists. Based on this lack of novelty and broader interest, I think this work is more suited to be published in specialised journals such as Organic Letters or Journal of Organic Chemistry.

Response: We thank the reviewer for this comment. One of the key unmet challenges in organic synthesis is selectivity. In the case of C-H functionalization, site-selectivity is essential. Despite that numerous C-H functionalization reactions are known, new ones are published regularly because many challenges associated with reactivity and selectivity exist. Although oxidation of simple methylarenes to aromatic aldehydes are known, site-selectivity remains an unsolved challenge. We believe what makes this work interesting is the site-selectivity in a complex setting. It would be rather difficult for one to predict the selectivity based on existing literature. This unexpected discovery is not limited to one type of benzoheterocycles. Benzoheterocycles are essential for many applications in chemistry and biology. The fact that non-aromatic heterocycles (45-51) are also selectively functionalized suggest that selectivity should be not limited to benzoheterocycles. Also note that this high selectivity is not limited to methoxylation

and is applicable to other benzylic C-H functionalization reactions using nucleophiles other than MeOH. Research in this direction are ongoing in our laboratory and will be published in due course. We believe that this discovery that perturbation of the electronics of the benzene ring can lead to unexpected high site selectivity in benzylic C-H functionalization would be of great interest to a broad readership.

Specific comments and questions are as follows:

1) Product 10 shows slight decrease in regioselectivity. How would regioselectivity be changed if stronger electron-withdrawing group such as tosyl or nosyl is attached? If the regioselectivity further decreases, it strengthens the authors' arguments that regioselectivity of the oxidation is controlled by electronic environment of a substrate.

Response: We thank the reviewer for this comment. Unfortunately, sulfonyl substituted substrates are not stable under the conditions probably because of solvolysis.

2) Yields are usually moderate (50-70%) .What are the rest of the material?

Response: We thank the reviewer for this question. We did not obtain identifiable side products in most of the reactions. Besides benzylic functionalization, the substrates may also undergo competing C(aryl)-H functionalization with MeOH. Such a process would lead to heteroatom substitution, which reduces the oxidation potential of the compound and promotes further oxidation and decomposition into untraceable material.

3) I don't fully understand the purpose of NBS bromination (figure 6a). Do the authors want to claim that mechanism is likely different based on the observation of regioisomers in the case of bromination?

Response: We thank the reviewer for this question. NBS bromination is known to proceed through hydrogen atom abstraction. The NBS bromination experiment was conducted to show that the electrochemical oxidation reaction did not proceed through hydrogen atom abstraction, which is a common mechanism for C-H functionalization reactions.

4) Analysis of LUMO of radical cation to account for the regioselectivity makes sense, but I found it is not very intuitive. Do you think we can draw the same conclusion by using the stability of benzylic radicals, instead of LUMO of radical cation?

Response: We thank the reviewer for this question. Taking C4,C6-Me substrates for

example, because of the benzylic positions are on the same benzene ring in 1,3-position, it is not easy to judge the relative stability of the benzyl radicals because resonance distribution of radicals is the same for both radicals on C2, C4 and C6. As a result, it is not intuitive that these reactions would be very selective. We believe these nonintuitive findings are what makes the work interesting. On the other hand, LUMO and NPA charges clearly showed the difference.

Reviewer #3 (Remarks to the Author):

In this communication, Cheng, Xu and co-workers reported a method for site-selective electrochemical oxidation. The benzylic methyl C-H bond is cleaved via anodically oxidation to afford the coupling product. The structure of the synthetic products were confirmed through X-ray single crystal analysis. The site-selective product may be subjected to hydrolysis, affording the corresponding aldehydes, which could be converted into the antihypertensive drug telmisartan. Additional DFT calculations have been carried out to explore the detailed mechanism of this transformation. The reaction has been demonstrated on various heterocycles with excellent yields and excellent site-selectivity. However, the manuscript and the supplementary information lack any detailed disclosure on the proposal mechanistic cycle. This should be remedied if possible.

Response: We thank the reviewer for the comment. A proposed mechanism has been added to the Supporting Information.

Overall, this transformation represents a useful method to access aromatic aldehydes from methylarenes. The substrates scope was broad with great regiocontrol. The broad selection of substrates may elicit general interest in the community. Taken together, this manuscript does possess the broad impact expected of a Nat. Comm. Publication, but some revisions are necessary prior to publication (vide supra):

Response: We thank the reviewer for the favorable recommendation.

1. The scope of the reaction has been well demonstrated by the authors. Under optimized conditions, substrates reacted site-selectively at C6-Me group to give dimethyl acetal. However, in some case, site-selective oxidation of C4-Me group could also be achieved (40'). How about the other kinds of the substrate only with C4-Me group? This is a minor point, but if the authors have already attempted other substrates, it's better to include them, even if the results are not good.

Response: We thank the reviewer for this question. C4 and C6 Me groups are in a 1,3-position and are thus similar electronically. Without a C6 Me, C4 Me can be

oxidized as we can see from products 44 and 57'. We have not attempted substrates with only C4-Me. We instead have focused on the selectivity study of substrates with two or more benzylic positions. We do believe that it is worth exploring further the selectivity and reactivity on all the positions and are currently working on this direction. Further studies will be published in due course.

2. Aside from the foregoing, the reviewer has some concerns over the title of the manuscript. A wide range of structurally diverse aromatic dimethyl acetal derivatives has been disclosed via the electrooxidation of methyl heterocycles in a site-selective manner. However, the dimethyl acetals were isolated as the products. There is inaccurate in highlighting aldehyde in the title.

Response: We thank the reviewer for this comment. We have changed the title to "Site-selective electrooxidation of methylarenes to aryl acetals" and modified relevant description in the abstract and main text. We do recognize that we isolate aryl acetal as products instead of the aromatic aldehydes in most of the reactions. The conversion of acetals to aldehydes have been achieved by hydrolysis in a separate step as shown in Fig 5 during the synthesis of Telmisartan or by acidic aqueous workup as demonstrated for compound 39 and 52'. Acetal formation and hydrolysis has been employed in industrial scale production of p-anisaldehyde and in recently reported studies on aromatic aldehyde synthesis (Angew. Chem. Int. Ed. 2017, 56, 7191; Angew. Chem. Int. Ed. 2017, 56, 1500). These authors described their synthesis, acetal formation followed by hydrolysis, as formylation. But of course, we have only tried hydrolysis for few compounds.

Reviewer #4 (Remarks to the Author):

The paper entitled "Site-selective electrooxidation of methylarenes to aromatic aldehydes" focuses on the synthesis of acetals starting from methylarenes using electrochemical tools to be further used for aldehyde synthesis approaches. This work is an ongoing research in which the authors previously reported an electrochemical synthesis of functionalized benzimidazoles and imidazopyridines via amidine cyclization. For this new manuscript the authors obtained a very completed and exhaustive library which is quite well characterized with the suitable techniques for the purpose. The compounds were produced in an eco-friendly mode coupled with good side-selectivity and yields which is crucial for future scale-up prospective. Therefore, the following minor remarks must be addressed by the authors before this paper could be published in "Nature Communications" journal.

Response: We thank the reviewer for the favorable recommendation.

1. In fact, the authors synthesize acetals employing electrochemical tools with methylarenes as starting materials. However, the aromatic aldehydes were obtained under further hydrolysis of these acetals. Therefore, this article title is misleading, while someone is trying to search for a paper, the first impression this title causes it that the authors obtained aromatic aldehydes from electrooxidation techniques, which is not completely true (just the previous step, or two steps before the hydrolysis that give rise to the final aldehydes). In contrast, other papers such as *J. Electroanal. Chem.* 751 (2015) 105–110, which is entitled “A promising electro-oxidation of methyl-substituted aromatic compounds to aldehydes in aqueous imidazole ionic liquid solutions” is an example where the authors did electrochemical synthesis from methyl-substituted compounds to obtain the aldehydes as final products. Also, sentences such as referred in page 2: “Herein we report a generally applicable electrochemical strategy capable of oxidizing various methyl benzoheterocycles to aromatic aldehydes in a site-selective manner” shall be modified accordingly along the manuscript.

Response: We thank the reviewer for this comment. We have changed the title to “Site-selective electrooxidation of methylarenes to aryl acetals” and modified relevant description in the abstract and main text. We do recognize that we isolate aryl acetal as products instead of the aromatic aldehydes in most of the reactions. The conversion of acetals to aldehydes have been achieved by hydrolysis in a separate step as shown in Fig 5 during the synthesis of Telmisartan or by acidic aqueous workup as demonstrated for compound 39 and 52'. Acetal formation and hydrolysis has been employed in industrial scale production of p-anisaldehyde and in recently reported studies on aromatic aldehyde synthesis (*Angew. Chem. Int. Ed.* 2017, 56, 7191; *Angew. Chem. Int. Ed.* 2017, 56, 1500). These authors described their synthesis, acetal formation followed by hydrolysis, as formylation. But of course, we have only tried hydrolysis for few compounds.

2. At the end of introduction the authors shall insert a paragraph telling the methylarenes choice (you can also highlight that this is an ongoing research by the group in which previous steps like the benzimidazoles electrosynthesis via amidine cyclization were performed with success) and also that main focus is targeting compounds like the antihypertensive drug telmisartan in an eco-friendly path. So that, it is easier for the reader to follow the main aim of this research.

Response: We thank the reviewer for this suggestion. We have modified the introduction by following the suggestion.

3. After table 1, the authors are not talking about the reaction conditions anymore and

thereafter a simple sentence must be inserted here to separate the reaction conditions optimization “section” from the use of compounds with different substituents, so as to the readers can follow better the logical reasoning applied.

Response: We thank the reviewer for this suggestion. The following sentence has been added: With the optimized reaction conditions defined, we set out to explore the scope of the electrooxidation of methylenes.

4. No efficiency reduction was observed from entry 1 to entry 7 (Table 1), as it is referred in page 3, line 3, and therefore must be excluded from that sentence.

Response: We thank the reviewer for this suggestion. We have removed description of entry 7 from the sentence and described it separately.

5. Page 5, I agree with last sentence where the authors say that site-selectivity is not controlled by steric effects. However, the authors refer some lines above that “...resultant product mixture comprised mono-methoxylated 42 as the main product with a minor amount of acetal 43, because the steric environment was detrimental to the second C–H cleavage”, if so, how do you explain the contrast obtained for compound 46? The text must be modified accordingly.

Response: We thank the reviewer for this comment. For the purpose of discussion, we number the starting material for 46 as 46S. The electrolysis of 46S can give the monomethoxy product 46' selectively in 74% yield if we use only half amount of time (electricity) as that of 46. This result along with some discussion has been included in the revised manuscript. For other substrates that are not sterically hindered, it is not possible to stop at monomethoxylation because monomethylated product is oxidized at close potential with the starting material. For the reaction of 46S, prolonged electrolysis led eventually to 46. But for 42, attempt to convert it completely to 43 by prolonging electrolysis resulted in decomposition of 42 and 43. Please note that radical cations are highly reactive species and can participate other reactions in addition to benzylic deprotonation, such as reaction with MeOH. The benzylic deprotonation is affected by the electronics and sterics of the arene. The deprotonation is most favored when the C-H bond and the π system are collinear, allowing the best orbital overlap for intramolecular ET between the C-H σ orbital and the SOMO in the ring, required for the C-H bond cleavage. This stereoelectronic requirement is hampered by the neighboring groups. The sterics become more severe after the introduction of the first OMe group, explaining the formation of monomethoxylation products 43 and 46'. Probably because of the low efficiency in deprotonation, the current efficiency for the formation of 46 is low (9.9 F/mol, 4 F/mol is needed in theory). In comparison, the current efficiency for formation of less

hindered 45 was 4.5 F/mol. Current efficiencies for each product are included in the Supporting Information.

6. Conclusions shall be improved, this article has impact and will get attention from several readers even more in a journal as "Nature Communications". The manuscript contains important matter for a scale-up prospective and I suggest the authors to refer something related with an industrial future point of view in order to avoid at a large scale the use of chemical oxidants and so on, as referred in the introduction.

Response: We thank the reviewer for this comment. We have modified the conclusion to include the points suggested by the reviewer.

REVIEWERS' COMMENTS:

Reviewer #1 (Remarks to the Author):

As the overall results of the reaction principle are certainly useful and all the raised comments have been addressed, I recommend it for the publication in Nature Communications.

Reviewer #2 (Remarks to the Author):

In the revised manuscript, most questions raised by reviewers were appropriately addressed. However, I consider the novelty and generality of this work are still questionable. Here are the reasons.

1) The authors responded to reviewer 2 that "One of the key unmet challenges in organic synthesis is selectivity. In the case of C-H functionalization, site-selectivity is essential. Despite that numerous C-H functionalization reactions are known, new ones are published regularly because many challenges associated with reactivity and selectivity exist. Although oxidation of simple methylarenes to aromatic aldehydes are known, site-selectivity remains an unsolved challenge. We believe what makes this work interesting is the site-selectivity in a complex setting." I agree with this response, but my concern is the selectivity in this work comes from electronic bias in a substrate. Substrate-controlled site-selectivity is not considered to be "development" to solve unmet challenges. To strengthen this argument, I would like to point out that substrate-controlled site-selective oxidation is rather common phenomenon. For example, ref10 in the manuscript also describes oxidation of alkylarenes to arylaldehydes via SET, where product 6f was clearly formed by selective oxidation among multiple methyl groups. What constitutes "novelty" in this context should be catalyst-controlled site-selectivity, which is not the case of this work.

2) Although some reviewers agreed that the reaction has broad substrate scope, this needs to be further examined because around 80% of the current scope is constituted of benzimidazole and benzothiazole. As the title indicates that this method is for methylarenes in general, the substrate scope should include broader range of methylarenes. For example, it might be interesting to use substrates that give mixture of regioisomers under known conditions such as 5s or 5w in ref10, and the starting material for product 21 in ref9.

Reviewer #3 (Remarks to the Author):

In the Xu's revised manuscript, the additional proposed mechanism discussion significantly approve the quality of this paper. The modifications in the text/SI and correspondence in the cover letter address all my concerns. This reviewer recommends this manuscript for Nat. Comm. publication.

Point-by-point response to the referees' comments is as follows:

REVIEWERS' COMMENTS:

Reviewer #1 (Remarks to the Author):

As the overall results of the reaction principle are certainly useful and all the raised comments have been addressed, I recommend it for the publication in Nature Communications.

Response: We thank the reviewer for the recommendation.

Reviewer #2 (Remarks to the Author):

In the revised manuscript, most questions raised by reviewers were appropriately addressed. However, I consider the novelty and generality of this work are still questionable. Here are the reasons.

1) The authors responded to reviewer 2 that "One of the key unmet challenges in organic synthesis is selectivity. In the case of C-H functionalization, site-selectivity is essential. Despite that numerous C-H functionalization reactions are known, new ones are published regularly because many challenges associated with reactivity and selectivity exist. Although oxidation of simple methylarenes to aromatic aldehydes are known, site-selectivity remains an unsolved challenge. We believe what makes this work interesting is the site-selectivity in a complex setting." I agree with this response, but my concern is the selectivity in this work comes from electronic bias in a substrate. Substrate-controlled site-selectivity is not considered to be "development" to solve unmet challenges. To strengthen this argument, I would like to point out that substrate-controlled site-selective oxidation is rather common phenomenon. For example, ref10 in the manuscript also describes oxidation of alkylarenes to arylaldehydes via SET, where product 6f was clearly formed by selective oxidation among multiple methyl groups. What constitutes "novelty" in this context should be catalyst-controlled site-selectivity, which is not the case of this work.

2) Although some reviewers agreed that the reaction has broad substrate scope, this needs to be further examined because around 80% of the current scope is constituted of benzimidazole and benzothiazole. As the title indicates that this method is for methylarenes in general, the substrate scope should include broader range of methylarenes. For example, it might be interesting to use substrates that give mixture of regioisomers under known conditions such as 5s or 5w in ref10, and the starting material for product 21 in ref9.

Response: We thank the reviewer for taking time to review the revised manuscript and providing valuable comments. We put our focus on the benzoheterocycles because these structures are prevalent in bioactive compounds and their oxidation are difficult for existing methods. As shown in Fig. 5b, reported methods including those of ref 9 and 10 are all inefficient for the oxidation of benzimidazole 75. The starting material for product 21 in ref 9 reacted regioselectively (r.r.>20:1) under our electrochemical conditions to give the desired acetal 54 in 61% yield (Fig. 3). In addition, we have also tested two other alkoxybenzenes bearing two methyl groups at 2,4- or 3,4-positions. Both reacted regioselectively (r.r.>20:1) at the position 4 Me group to give the corresponding acetals 52 and 53 (Fig. 3). These results suggest that our method is not limited to benzoheterocycles. More detailed investigation on oxidation of non-benzoheterocycles is a subject of ongoing research and will be reported independently in due course. Like other methods, our method also has its limitations. For examples, boronic acid 5s (compound 55 in Fig. 3) or trifluoroborate 5w from ref 10 failed under our electrochemical conditions. In addition, electron-deficient substrates with oxidation potentials exceed the decomposition potential of methanol solvent are also not suitable. For examples, while products 67-70 are formed successfully, compound 71 failed to undergo further Me oxidation because of competitive anodic oxidation of MeOH. These additional results have been added to Fig. 3 and Fig. 4.

Reviewer #3 (Remarks to the Author):

In the Xu's revised manuscript, the additional proposed mechanism discussion significantly approve the quality of this paper. The modifications in the text/Sl and correspondence in the cover letter address all my concerns. This reviewer recommends this manuscript for Nat. Comm. publication.

Response: We thank the reviewer for the recommendation.